# Evaluation of STANDARD™ M10 SARS-CoV-2, a Novel Cartridge-Based Real-Time PCR Assay for the Rapid Identification of Severe Acute Respiratory Syndrome Coronavirus 2

Laura Grumiro [1,*], Martina Brandolini [1], Giulia Gatti [1], Agata Scalcione [1], Francesca Taddei [1], Giorgio Dirani [1], Andrea Mancini [1], Agnese Denicolò [1], Martina Manera [1], Silvia Zannoli [1], Maria Michela Marino [1], Manuela Morotti [1], Valentina Arfilli [1], Arianna Battisti [1], Monica Cricca [1,2] and Vittorio Sambri [1,2]

[1] Unit of Microbiology, The Greater Romagna Hub Laboratory, 47522 Pievesestina, Italy
[2] Department of Experimental, Diagnostic and Specialty Medicine (DIMES), Alma Mater Studiorum—University of Bologna, 40138 Bologna, Italy
* Correspondence: laura.grumiro@auslromagna.it

**Abstract:** Since the beginning of the pandemic, SARS-CoV-2 has caused problems for all of world's population, not only in terms of deaths but also in terms of overloading healthcare facilities in all countries. Diagnosis is one of the key aspects of controlling the spread of SARS-CoV-2, and among the current molecular techniques, real-time PCR is considered as the gold standard. The availability of tests that allow for the rapid and accurate identification of SARS-CoV-2 is therefore of considerable importance. Moreover, if these tests allow for even minimal intervention by the operator, any risk of contamination is reduced. In this study, the performances of the new STANDARD™ M10 SARS-CoV-2 (SD Biosensor Inc., Suwon, Korea) rapid molecular test, which incorporates the above-mentioned features, were characterized. The clinical and analytical performances measured by testing different variants circulating in Italy of STANDARD™ M10 SARS-CoV-2 were compared to the test already on the market and recognized as the gold standard: Xpert Xpress SARS-CoV-2 (Cepheid, Sunnyvale, CA, USA). The results obtained between the two tests are largely comparable, suggesting that STANDARD™ M10 SARS-CoV-2 can be used with excellent results in the fight against the global spread of SARS-CoV-2.

**Keywords:** SARS-CoV-2; COVID-19; diagnostic testing; STANDARD™ M10 SARS-CoV-2; cartridge-based test; real-time PCR

## 1. Introduction

Coronavirus disease 2019 (COVID-19), which was caused by severe acute respiratory syndrome coronavirus 2 (SARS-CoV-2), arose in Wuhan (Hebei Province, China) in late 2019 [1,2]. The subsequent worldwide spread of SARS-CoV-2 has caused 613,410,796 confirmed cases of COVID-19, including 6,518,749 deaths reported by the WHO as of late September 2022 (https://covid19.who.int/consulted on 28 September 2022) [3].

SARS-CoV-2 is characterized by a 100 nm-long single-strand RNA genome that encodes 7 viral proteins and adheres to the host cell by means of Spike glycoprotein (S protein), resulting in the entry of the virus into the cell [4]. SARS-CoV-2 uses human angiotensin converting enzyme 2 (ACE2) as the key cellular receptor to enter cells [5]. Structures of the receptor-binding domain (RBD) of SARS-CoV-2 S protein in complex with ACE2 were determined in the early stage of the epidemic [6], showing how ACE2 might mediate SARS-CoV-2 entry within cells and explaining SARS-CoV-2 higher affinity with ACE2 compared to SARS-CoV-1.

The persistent chain of contagion and the global spread of the virus have facilitated the surge of a large number of mutations in the genome of SARS-CoV-2 with an increase in their accumulation over time, despite the virus itself having a quite low spontaneous mutation rate ($1.1 \times 10^{-3}$ per year) [7,8].

Most of SARS-CoV-2 genome modifications (deletions, insertions, and base substitutions) are silent mutations, therefore not affecting viral properties. Mutations that bring a selective advantage in terms of infection and pathogenicity are the most dangerous from an epidemiological point of view, forcing health authorities to undertake strict preventive social measures. In particular, mutations of the greatest concern involve the S gene, which encodes spike glycoprotein, the site of attack of neutralizing antibodies (NAb) [7,9].

Given the characteristics of SARS-CoV-2, it is therefore not surprising that in the last two years the RNA of the original Wuhan virus changed several times and new variants of concern have arisen worldwide [10]. In particular, across the end of 2020 and the first months of 2021, three notable variants of concern named Alpha (B.1.1.7), Beta (B.1.351), and Gamma (P.1) have been identified in three different countries (the UK, South Africa, and Brazil, respectively) [11]. The Delta (B.1.617.2) variant, first detected in India in late 2020, had already outpaced the other abovementioned variants by late spring 2021, on a global level [12]. Finally, in November 2021 the Omicron variant (B.1.1.529) was first identified in Botswana and in a few weeks overcame Delta globally [13]. Moreover, a great number of minor local mutations appeared in specific areas, without spreading on a large scale [14].

In this context, a crucial role is represented by SARS-CoV-2 diagnosis [15], and among the plethora of techniques used for diagnosis the gold standard is represented by reverse transcription PCR (RT-PCR), which can be used on different clinical specimens [16,17]. Since the beginning of the pandemic, the most common specimen types assayed have been nasopharyngeal (NP) and/or oropharyngeal (OP) swabs [18].

In this study, a new cartridge-based RT-PCR molecular test named STANDARD[TM] M10 SARS-CoV-2, developed by the South Korean SD Biosensor Company, was evaluated. The comparator test chosen for this study was Xpert[®] Xpress SARS-CoV-2 (Cepheid, Sunnyvale, CA, USA), due to its worldwide diffusion and similar technology [19].

Both devices are real-time cartridge-based tests and are characterized by minimum intervention by an operator [19]: STANDARD[TM] M10 and GeneXpert are both closed systems sharing the same philosophy: the modular instrument, which allows for random access, and continuous loading, and are both easy to use.

The key features of each evaluated test and instrument are summarized in Table 1.

**Table 1.** Comparison of key characteristics of the two tests used in the study.

| Characteristic | STANDARD[TM] M10 SARS-CoV-2 | Xpert[®] Xpress SARS-CoV-2 |
|---|---|---|
| Instrument | STANDARD M10 | GeneXpert |
| Specimen used in the study [1] | NP | NP |
| Test duration [2] | 60 min | 50 min |
| Loading | Random access | Random access |
| Genes detected | orf1Ab, E | E, N |

[1] NP = nasopharyngeal swab. [2] For a negative result. Early call for positive samples may occur earlier.

The aim of this study was, firstly, a comparative head-to-head analysis on routine samples, carried out on both systems to calculate the clinical sensitivity, specificity, negative predictive value (NPV), positive predictive value (PPV), and accuracy of STANDARD[TM] M10 SARS-CoV-2 compared to Xpert[®] Xpress SARS-CoV-2.

Secondly, the analytical sensitivity of both tests was calculated on five different variants and one subvariant, namely, Beta; Gamma; Delta (including subvariant Delta plus); Omicron; and Bagnacavallo, a local variant identified by our laboratory in the surrounding areas [14]. Control material at a known specific concentration was used as a reference.

## 2. Materials and Methods

### 2.1. Specimen Collection and Storage

A number of 195 fresh specimens, including 99 positive and 96 negative specimens, were detected parallelly by Xpert Xpress SARS-CoV-2 (Cepheid, Sunnyvale, CA, USA) and the STANDARD[TM] M10 SARS-CoV-2 (SD Biosensor Inc., Suwon, Korea).

The samples used in this study do not belong to any particular category; they were collected randomly among all the samples tested in our laboratory during the period in which the study was conducted. The clinical features of the patients and the severity of the symptoms are not known. Nasopharyngeal nylon swabs of test subjects were performed by qualified personnel and transported in 3-mL universal transport medium (UTM) tubes (Copan UTM®, Copan, Italy), to be processed according to the guidelines in force. After routine detection, samples were aliquoted and stored at −80 °C immediately.

### 2.2. Cepheid Xpert Xpress SARS-CoV-2 Assay

For each testing, 300 μL of the nasopharyngeal swab specimen was added into the cartridge sample chamber using a transfer pipette. After closing the lid, the cartridge was loaded onto the GeneXpert instrument for automated real-time viral RNA extraction and detection. If the SARS-CoV-2 signal for the N2 nucleic acid target or signals for both nucleic acid targets (N2 and E) had Ct values within the valid range (<45) and end points above the minimum setting, this was considered a positive result. If the SARS-CoV-2 signal for only the E nucleic acid target had a Ct within the valid range and an end point above the minimum setting, the result was considered presumptive positive. Given the Italian epidemiological context, the presence of a presumptive positive result was considered a positive since it was highly indicative of the presence of SARS-CoV-2. If the SARS-CoV-2 signals for the two nucleic acid targets (N2 and E) did not have a Ct within the valid range and an end point above the minimum setting, the result was considered negative.

### 2.3. STANDARD M10 SARS-CoV-2 Assay

For each testing, 600 μL of UTM nasopharyngeal sample was added, after vortexing, to the cartridge's sample chamber. The cartridge was loaded onto the STANDARD[TM] M10 instrument (SD Biosensor Inc., Suwon, South Korea) for automated real-time RNA detection. The amplified nucleic acid targets were ORF1ab and E. Positive results were given if ORF1ab or both targets had Ct values within the valid range (<38), and endpoints above the minimum setting. Similarly to the previously described GeneXpert System, if only E gene showed Ct within the range and endpoint above the minimum, the results were considered presumptive positive, whereas if none of the targets met the criteria, the sample results were considered negative. Additionally, in this case, given the Italian epidemiological context, the presence of a presumptive positive result is considered a positive since it is highly indicative of the presence of SARS-CoV-2.

### 2.4. Limit of Detection (LoD) Calculation on Different SARS-CoV-2 Variants

A commercial standard SARS-CoV-2 RNA, namely, AccuPlex SARS-CoV-2 Molecular Controls Kit-Full Genome (catalog number PN 0505-0159), was used for a limit-of-detection (LOD) study. The concentration of the standard was 5000 cp/mL, which was then diluted in buffer phosphate (pH 7.4) to 400 cp/mL (1:12.5 fold), and subsequently 2-fold dilution has been applied till the final concentration of 25 cp/mL.

For clinical samples belonging to different variants, dilution was carried out directly by the stocked samples stored at −80 °C following the procedure indicated in Brandolini et al. 2021 [20]. Briefly, the initial concentration of the sample has been determined and dilution has been made in buffer phosphate in order to test the concentration range between 20,000 and 1250 cp/mL.

Clinical samples belonging to clinical variants were tested 1 time at 5 different concentrations. The variants tested were Beta (B.1.351), Gamma (P.1), Delta (B.1.617.2), Bagnacavallo (B.1.1.7 + Δ619_624 N gene), Omicron (B.1.1.529), and the subvariant Delta plus

(B.1.617.2 AY.4.2). Lineage was determined by whole-genome next-generation sequencing with Illumina (Illumina Inc., San Diego, CA, USA) and assigned according to the Phylogenetic Assignment of Named Global Outbreak LINeages (Pangolin).

## 3. Results

### 3.1. Comparative Analysis STANDARD^TM M10 SARS-CoV-2 vs. Xpert® Xpress SARS-CoV-2

Of the 195 prospective NP swab specimens, 193 yielded the same result on both platforms, resulting in an agreement of 98.97% (193/195).

Both tests include the E gene as a target. When the amplification of only the E gene happens, both tests provide the result "presumptive positive". This particular case happened three times with STANDARD^TM M10 and once for Xpert® Xpress SARS-CoV-2. Samples characterized by the only amplification of the E gene have been considered positive since the amplification of the E gene from Sarbecovirus, given the Italian epidemiological situation, is highly indicative of the presence of SARS-CoV-2.

Ninety-seven specimens were detected by both tests (STANDARD^TM M10 SARS-CoV-2 positive/Xpert® Xpress SARS-CoV-2 positive), two were detected positive only by Xpert® Xpress SARS-CoV-2 (STANDARD^TM M10 SARS-CoV-2 negative/Xpert® Xpress SARS-CoV-2 positive), and 96 were detected negative by both tests (STANDARD^TM M10 SARS-CoV-2 negative/Xpert® Xpress SARS-CoV-2 negative). These results allow one to calculate sensitivity (97.98%), specificity (100%), PPV (100%), and NPV (97.96%) for STANDARD^TM M10 SARS-CoV-2 (Table 2). The overall agreement between the two tests (accuracy) is 98.97%.

**Table 2.** Results obtained comparing STANDARD^TM M10 SARS-CoV-2 vs. Xpert® Xpress SARS-CoV-2.

| STANDARD^TM M10 SARS-CoV-2 | | | | | | | |
|---|---|---|---|---|---|---|---|
| | | Pos | Neg | Tot | Sensitivity | 97.98% | |
| Xpert® Xpress SARS-CoV-2 | Pos | 97 | 2 | 99 | Specificity | 100% | |
| | Neg | 0 | 96 | 96 | PPV | 100% | |
| | Tot | 97 | 98 | 195 | NPV | 97.96% | |

Cohen's kappa analysis comparing the performances of the two tests determined a k value of 0.98, which indicates a rate of "almost perfect" [21].

By analyzing the data of two discordant samples (from a total of 195), the N2 Cycle threshold (Ct) values obtained with Xpert Xpress were 37, and both samples were considered positive; the E gene was identified only once in 35 samples. The Ct values of the two discordant sample indicate a low viral load.

To further assess the agreement between the two platforms, we analyzed the relationship between the E gene detected on each system. Ct values exclusively for the STANDARD^TM M10 SARS-CoV-2 E-gene target ranged from 10.1 to 33.3, while for the Xpert Xpress SARS-CoV-2 E-gene, detection ranged from 13.2 to 37. In 81 out of 95 cases, STANDARD^TM M10 SARS-CoV-2 detected an earlier Ct value than Xpert Xpress SARS-CoV-2; however, the differences between the Ct values were low, 1.3.

### 3.2. Analytical Sensitivity on Different Variants

Five different variants, namely, Beta (B.1.351), Gamma (P.1), Delta (B.1.617.2), Bagnacavallo (B.1.1.7 + Δ619_624 N gene), Omicron (B.1.1.529), and the subvariant Delta plus (B.1.617.2 AY.4.2) were tested with both STANDARD^TM M10 SARS-CoV-2 and Xpert® Xpress SARS-CoV-2. The concentration tested ranged from 20,000 cp/mL to 1250 cp/mL. The AccuPlex SARS-CoV-2 Molecular Controls Kit was used as a reference.

STANDARD^TM M10 SARS-CoV-2 was less sensitive than Xpert® Xpress SARS-CoV-2 on Gamma (P.1) variant and Delta plus subvariant (B.1.617.2 AY.4.2), with an estimated LoD of 2500 cp/mL and 5000 cp/mL, respectively, while Xpert® Xpress SARS-CoV-2 showed an LoD of 1250 cp/mL on the Gamma variant and 2500 cp/mL on the Delta plus subvariant.

Regarding other variants, including the circulating ones (e.g., Omicron), the results are comparable, with an LoD of 1250 cp/mL (Table 3).

**Table 3.** Comparative analysis of STANDARD<sup>TM</sup> M10 SARS-CoV-2 vs. Xpert® Xpress SARS-CoV-2 on different variants circulating in Italy. AccuPlex SARS-CoV-2 Molecular Controls Kit was used as reference material.

| Variant/Subvariant Name | Concentration (cp/mL) | STANDARDTM M10 SARS-CoV-2 | | Xpert®Xpress SARS-CoV-2 | |
|---|---|---|---|---|---|
| | | E Gene (Ct) * | Orf1ab Gene (Ct) * | E Gene (Ct) * | N2 Gene (Ct) * |
| B.1.351 (BETA) | 20,000 | 30.02 | 29.70 | 31.2 | 34.9 |
| | 10,000 | 30.88 | 30.04 | 32.5 | 36.5 |
| | 5000 | 31.95 | 32.53 | 32.4 | 35.4 |
| | 2500 | 33.56 | 34.13 | 35.1 | 38.6 |
| | 1250 | 32.93 | N/A | 40.4 | 39.4 |
| P.1 (GAMMA) | 20,000 | 31.28 | 31.58 | 32.9 | 36.5 |
| | 10,000 | 31.99 | 32.50 | 33.6 | 37.6 |
| | 5000 | 34.27 | 33.47 | 34.1 | 38.2 |
| | 2500 | 33.65 | 33.75 | 39.8 | N/A |
| | 1250 | N/A | N/A | 39.8 | 41.8 |
| B.1.617.2 (DELTA) | 20,000 | 31.22 | 30.91 | 34 | 36.9 |
| | 10,000 | 34.37 | 33.62 | 33.7 | 37.3 |
| | 5000 | 34.77 | N/A | 36 | 38.4 |
| | 2500 | 33.86 | 34.53 | 39.5 | 40.8 |
| | 1250 | 35.43 | N/A | 38.3 | 41.8 |
| B.1.617.2 AY.4.2 (DELTA PLUS) | 20,000 | 33.02 | 33.37 | 34.5 | 37.6 |
| | 10,000 | 34.14 | N/A | 36.1 | 39.1 |
| | 5000 | 34.45 | N/A | 38 | 40 |
| | 2500 | N/A | N/A | 42.8 | 42.3 |
| | 1250 | N/A | N/A | N/A | N/A |
| B.1.1.7 + Δ619_624 N gene (BAGNACAVALLO) | 20,000 | 31.24 | 30.80 | 33.4 | 36.4 |
| | 10,000 | 32.69 | 32.95 | 36 | 38 |
| | 5000 | 33.19 | 32.39 | 37.4 | 40.1 |
| | 2500 | N/A | 34.15 | 39.6 | 41.5 |
| | 1250 | 34.22 | 34.31 | 38.3 | 41.4 |
| B.1.1.529 (OMICRON) | 20,000 | 30.35 | 32.61 | 32 | 34.1 |
| | 10,000 | 31.96 | 31.71 | 33.6 | 35.8 |
| | 5000 | 31.88 | 33.6 | 33.8 | 36 |
| | 2500 | 33.33 | N/A | 35.8 | 38.3 |
| | 1250 | 32.34 | N/A | 35.2 | 37.8 |
| AccuPlex SARS-CoV-2 Molecular Controls Kit | 400 | 32.55 | 31.96 | 34.1 | 37.2 |
| | 200 | 32.59 | N/A | 36.7 | 40.3 |
| | 100 | 34.95 | N/A | 36.8 | 39.4 |
| | 50 | N/A | N/A | N/A | 41 |
| | 25 | N/A | N/A | 40.2 | 41.5 |

* N/A not applicable.

Interestingly, testing standard-RNA-concentration controls, the resulting LoD appeared lower, in particular, STANDARD<sup>TM</sup> M10 SARS-CoV-2 reached the declared one of 100 cp/mL [22], and Xpert® Xpress SARS-CoV-2 reached 25 cp/mL (Table 3).

STANDARD<sup>TM</sup> M10 SARS-CoV-2 identifies with more efficiency E gene rather than ORF1ab.

## 4. Discussion

For two years, SARS-CoV-2 has challenged health care systems around the world. The development of vaccines has limited the serious effects of the related disease, but on the other hand the variability and mutability of SARS-CoV-2 has posed and still poses a further challenge [23,24]. In particular, SARS-CoV-2 mutations are able to circumvent the

protection of vaccines in terms of virus spread, and the succession of different variants of concern over the last two years draws attention to this aspect [25,26]. It should also be remembered that the ability of SARS-CoV-2 to mutate is not surprising given that the 'old' SARS-CoV-1 also showed moderate mutation rates [27]. In this context, the role of diagnosis is crucial. Having molecular methods available that are able to detect the virus independently of any mutations is fundamental because the first step towards containing the virus is diagnostic identification [28]. If identification happens in a quick and accurate manner, this is surely a plus [29].

In this work, we evaluated the performance of the new STANDARD[TM] SARS-CoV-2 rapid molecular test from the South Korean company SD Biosensor. The test was compared with Cepheid's Xpert Xpress SARS-CoV-2 for two main reasons:

1. Similarity—both tests are encapsulated in disposable cartridges that allow for the extraction of viral RNA and the subsequent amplification and identification by RT-PCR without any operator intervention other than loading the sample into the cartridge. This ensures operator safety and minimizes the possibility of cross-contamination between samples. Both tests are performed using modular instruments: STANDARD[TM] M10 in the case of STANDARD[TM] SARS-CoV-2 and GeneXpert in the case of Xpert Xpress SARS-CoV-2. Both instruments are characterized by random access and continuous loading. Responses are rapid (less than 60 min) and allow the clinician to make the best decisions for patient management.

2. Xpert Xpress SARS-CoV-2 is a widely used test worldwide and, despite the fact that all like tests on the market can be affected by possible virus mutations [30], it is considered one of the best-performing tests on the market [19,29].

As shown in the results section, the performance of STANDARD[TM] M10 SARS-CoV-2 is largely superimposable with that of Xpert Xpress SARS-CoV-2. In particular, out of 195 samples tested, 193 were concordant with both instruments.

The two discrepant samples, i.e., those identified as positive with Xpert Xpress SARS-CoV-2 and negative with STANDARD[TM] M10 SARS-CoV-2, were characterized by Ct close to the declared cut-off (38 ct) of STANDARD[TM] 10 SARS CoV-2 [22]. In particular, one sample was characterized by the exclusive presence of the N2 gene and the other by both genes. A possible explanation for these two discrepant results is the low viral load of the two samples, which was only identified with Xpert Xpress SARS-CoV-2. In particular, the case with exclusive amplification of N2 target at high Ct (>35) can be considered a negative result [31]. It is worth noting that out of 97 positive samples (excluding the two discrepancies just mentioned), 14 had Ct values for the N2 gene between and 41 with Xpert Xpress SARS-CoV-2 (Supplementary Table S1). In all of these cases, STANDARD[TM] M10 identified the samples as positive and showed good performance even with low charge samples.

Both tests, STANDARD[TM] 10 SARS-CoV-2 and Xpert Xpress SARS-CoV-2, were able to identify the main variants circulating in Italy, including the local variant (Table 3). The Xpert[®] Xpress SARS-CoV-2 test was slightly more sensitive than the STANDARD[TM] M10 SARS-CoV-2 test in that it was able to identify SARS-CoV-2 at a lower dilution in the case of the Gamma variant and the Delta plus sub-variant (Table 3). This trend was also confirmed with a control of known concentration; in this case, STANDARD[TM] 10 SARS-CoV-2 reached the claimed LoD of 100 cp/mL, while Xpert Xpress SARS-CoV-2 identified RNA up to 25 cp/mL. Analysis of the data in Table 3 shows that the clinical samples were identified at higher concentrations than the control, with the reason for this difference being the nature of the clinical samples. It is possible that even though the initial concentration of the samples was calculated using the algorithm described in Brandolini et al., there was a slight overestimation of the initial sample (the initial samples had concentrations of $3 \times 10^{10}$ viruses) [20]. On the other hand, by using a control characterized by a specific concentration of 5000 cp/mL, any dilution errors were eliminated.

## 5. Conclusions

A final thought on the different sensitivity of the two methods under study: in the light of worldwide epidemiology, it is important to understand the clinical significance of high analytical sensitivity. Cevik and colleagues have shown that viral RNA can persist in patients for several days if not weeks, but this persistence does not correlate with the patients' ability to infect; in particular, prolonged RNA shedding correlates with the presence of a non-viable virus [32–34]. It is therefore very important to know not who is a virus carrier but who is capable of spreading an infectious virus; various literature data show how high ct (ct > 38) generally correlates with people who do not spread a viable virus [35,36]. The best solution to this challenge is not to rely on Ct alone but to correlate it with the patients' symptoms [37]. In this context, having a test such as STANDARD^TM M10 SARS-CoV-2, which is characterized by a cut-off at ct 38, together with other laboratory tests, can be of great help in understanding who is really infected with viable virus. More generally, this work shows the good performance of STANDARD^TM M10 SARS-CoV-2 with routine clinical samples, and for the main variants circulating in Italy, the use of a PoC test that provides fast and accurate results can be a valuable aid to control the spread of SARS-CoV-2 in different healthcare settings.

**Supplementary Materials:** The following supporting information can be downloaded at: https://www.mdpi.com/article/10.3390/applmicrobiol2040067/s1, Table S1: Complete dataset about the comparative analysis of STANDARD^TM M10 SARS-CoV-2 vs. Xpert® Xpress SARS-CoV-2.

**Author Contributions:** Conceptualization, V.S.; formal analysis, L.G.; data curation, L.G., M.B., G.G., A.S., F.T., G.D., A.M., A.D., M.M. (Martina Manera), S.Z., M.M.M., M.M. (Manuela Morotti), V.A. and A.B.; writing—original draft preparation, L.G.; writing—review and editing, L.G. and M.B.; validation, G.D., M.C. and V.S.; supervision, G.D. and V.S.; and project administration, V.S. All authors have read and agreed to the published version of the manuscript.

**Funding:** This research received no external funding.

**Institutional Review Board Statement:** The study was conducted according to the guidelines of the Declaration of Helsinki and approved by the Institutional Review Board of AUSL ROMAGNA (protocol code "COVdPCR" of 7 February 2020).

**Informed Consent Statement:** The samples included in this study were sent to the Unit of Microbiology, Greater Romagna Area Hub Laboratory, Cesena, Italy for routine diagnostic purposes, and the laboratory data results were reported as an answer to a clinical suspicion. As such, informed consent from patients was not required. Prior to testing, all samples underwent the anonymization procedure used at the Unit of Microbiology of the Hub Laboratory of the Great Romagna Area, in order to adhere to the regulations issued by the local Ethical Board (AVR-PPC P09, rev.2; based on Burnett et al., 2007) [38].

**Data Availability Statement:** The data presented in this study are available on request from the corresponding authors.

**Acknowledgments:** The authors wish to thank Relab srl for providing the STANDARD^TM M10 cartridges and AccuPlex SARS-CoV-2 Molecular Controls Kit. In addition, the authors wish to thank Davide Antoniani, Francesca Garbarino, and Silvia Vettore from Relab srl for the critical review of the manuscript and Alessandro Rossi from Relab srl for technical help.

**Conflicts of Interest:** The authors declare no conflict of interest.

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
