# Peer review of "Evaluation of STANDARDTM M10 SARS-CoV-2, a Novel Cartridge-Based Real-Time PCR Assay for the Rapid Identification of Severe Acute Respiratory Syndrome Coronavirus 2"

_2673-8007, doi:10.3390/applmicrobiol2040067_

Round 1
Reviewer 1 Report
Could the authors be so kind and comment regarding the severity of the 195 cases and also how did they decide who is going to enter which group of tests
Could the authors comment more on the importance in detecting the patients that will still remain positive and which of the two methods would be better to use in order to detect those carrier
Author Response
We thank Reviewer 1 very much for their comments about our work and we hope the changes we have made to the manuscript meet their suggestions.

Reviewer 2 Report
Laura Grumiro et al present a a special scientific work about evaluation of STANDARDTM M10 SARS-CoV-2, a novel cartridge-based Real Time PCR assay for the rapid identification of Severe Acute Respiratory Syndrome Coronavirus 2.
The article is very well written, and the authors deserve congratulations. From the clinician's point of view, I recommend that the conclusions be noted at the end of the article in a separate paragraph.
Author Response
We thank Reviewer 2 very much for their comments about our work and we hope the changes we have made to the manuscript meet their suggestions.
